# Angiotensin-converting enzyme 2 (ACE2) expression increases with age in patients requiring mechanical ventilation

Steven Andrew Baker[ID]*, Shirley Kwok, Gerald J. Berry[ID], Thomas J. Montine[ID]

Department of Pathology, Stanford University School of Medicine, Stanford, California, United States of America

* sab73@stanford.edu

**Data Availability Statement:** All relevant data are within the paper and its Supporting Information files.

## Abstract

Mortality due to Covid-19 is highly associated with advanced age, owing in large part to severe lower respiratory tract infection. SARS-CoV-2 utilizes the host ACE2 receptor for infection. Whether ACE2 abundance in the lung contributes to age-associated vulnerability is currently unknown. We set out to characterize the RNA and protein expression profiles of *ACE2* in aging human lung in the context of phenotypic parameters likely to affect lung physiology. Examining publicly available RNA sequencing data, we discovered that mechanical ventilation is a critical variable affecting lung *ACE2* levels. Therefore, we investigated ACE2 protein abundance in patients either requiring mechanical ventilation or spontaneously breathing. ACE2 distribution and expression were determined in archival lung samples by immunohistochemistry (IHC). Tissues were selected from the specimen inventory at a large teaching hospital collected between 2010–2020. Twelve samples were chosen from patients receiving mechanical ventilation for acute hypoxic respiratory failure (AHRF). Twenty samples were selected from patients not requiring ventilation. We compared samples across age, ranging from 40–83 years old in the ventilated cohort and 14–80 years old in the non-ventilated cohort. Within the alveolated parenchyma, ACE2 expression is predominantly observed in type II pneumocytes (or alveolar type II / AT2 cells) and alveolar macrophages. All 12 samples from our ventilated cohort showed histologic features of diffuse alveolar damage including reactive, proliferating AT2 cells. In these cases, ACE2 was strongly upregulated with age when normalized to lung area (p = 0.004) or cellularity (p = 0.003), associated with prominent expression in AT2 cells. In non-ventilated individuals, AT2 cell reactive changes were not observed and ACE2 expression did not change with age when normalized to lung area (p = 0.231) or cellularity (p = 0.349). In summary, ACE2 expression increases with age in the setting of alveolar damage observed in patients on mechanical ventilation, providing a potential mechanism for higher Covid-19 mortality in the elderly.

**Funding:** The author(s) received no specific funding for this work.

**Competing interests:** The authors have declared that no competing interests exist.

## Introduction

The emergence of severe acute respiratory syndrome coronavirus 2 (SARS-CoV-2) has led to a global pandemic of related illness grouped under the term coronavirus disease 2019 (Covid-19) [1]. Manifestations include both upper and lower respiratory infection as well as various non-respiratory organ system disturbances [2–6], including renal [7], gastrointestinal [8] and vascular abnormalities [9]. The vast majority of deaths result from lower respiratory tract infection leading to acute respiratory distress syndrome (ARDS) with the pathological findings of diffuse alveolar damage (DAD) [10, 11].

Risk factors for severe respiratory illness include hypertension [12], diabetes [12], obesity [13], cancer [14], chronic respiratory illness [15], male sex [13], and particularly advanced age [16–18]. For each year of age, the odds ratio of death has been estimated to increase by 1.1 in multivariable analysis [16]. Among individuals admitted to the ICU for Covid-19, risk of death for those $\geq$ 64 years old is 2.4-fold greater than those younger [19]. In contrast, mortality among children infected with SARS-CoV-2 is exceedingly rare compared to middle-aged individuals [20], a feature which is atypical for respiratory viruses [21]. The underlying reasons for this dramatic skew are currently unknown [22].

SARS-CoV-2 depends on the angiotensin-converting enzyme 2 (ACE2) protein for attachment to and infection of host cells [23]. ACE2 is a transmembrane metalloprotease belonging to the Renin-Angiotensin-Aldosterone System (RAAS) which processes angiotensin II into angiotensin 1–7 leading to vasodilation [24]. The gene encoding ACE2 (*Angiotensin I converting enzyme 2*, *Ace2*) exhibits strong RNA expression in the kidney, lung, bladder, stomach, small intestine, large intestine, and adipose tissue of rodents [25]. In rats, lung ACE2 protein expression has been shown to drastically decrease with age [26], raising the question of why older individuals are more susceptible to severe lower respiratory infection than the young [27]. Previous studies of *ACE2* levels in the human lung have revealed no significant change with respect to age in bulk RNA sequencing (RNAseq) analysis [28, 29]. Similar findings have been observed in single cell RNA (scRNA) expression data for *Ace2* in rodents [29]. In order to investigate this apparent paradox, we reexamined publicly available *ACE2* RNAseq data from human lungs [30], adjusting for various donor phenotypes. Our analyses reveal a significant increase in detected *ACE2* RNA levels associated with mechanical ventilation. After including the effect of mechanical ventilation, we find that *ACE2* expression increases with age in human lung samples from deceased donors. Although RNA levels often correlate with protein expression, exceptions are common [31, 32]. We sought to validate our findings at the RNA level by investigating ACE2 protein expression in archival human lung tissue from patients receiving supportive mechanical ventilation. Consistently, ACE2 protein expression strongly increased with age in patients requiring mechanical ventilation. These findings, at both the RNA and protein level, provide a potential explanation for the preponderance of severe Covid-19 cases among the elderly.

## Materials and methods

### RNAseq analysis

Normalized gene expression data and subject phenotypes from the NIH Genotype-Tissue Expression (GTEx) project version 8 (dbGaP accession phs000424.v8.p2, released 8/26/2019) were acquired from the GTEx Portal [33]. These data include 17,382 samples from 948 deceased human donors. Expression scores in Transcripts Per Kilobase Million (TPM) were extracted for *ACE2* from each sample and analyzed using R version 3.5.0. The expression for lung was plotted using ggplot2 stratified by donor age, sex, and/or the Hardy scale score (see **S**

**Methods** in **S1 File**), which are provided through the portal [33]. A linear model was fit using the *lm* function in base R to predict *ACE2* expression with age, sex, and Hardy score as covariates. Primary analyses were carried out on linear TPM data to facilitate interpretation. The significance and direction of the effects were confirmed after Box-Cox transformation of lung *ACE2* expression using $\lambda_{MLE}$ computed by the MASS package in R. Mouse scRNAseq data [34] for *Ace2* expression were obtained from the Tabula Muris portal [35] and droplet sequencing results for lung were plotted using ggplot2 in R.

## Immunohistochemistry (IHC)

All aspects of this study were approved by the Stanford Institutional Review Board under protocol #33727. Formalin-fixed paraffin-embedded (FFPE) human lung excisions/biopsies received through Stanford Surgical Pathology during 2010–2020. To allow for blinding, cases were chosen by 1 investigator who did not participate in specimen preparation or data acquisition and analyses until all data had been collected. We identified a set of 20 patients who were not on mechanical ventilation (other than for procedural anesthesia) and a set of 11 patients who required supportive ventilation for acute hypoxic respiratory failure (AHRF); either idiopathic or in the setting of advanced fibrotic interstitial lung disease. Patients in our non-ventilated cohort ranged from 14 to 80 years of age (mean 41.6 +/- 23.1 years). Patients in our ventilated cohort ranged from 40 to 83 years old (mean 56.8 +/- 12.3 years). Sections were stained using standard procedures (see **S Methods** in **S1 File**) with a commercial antibody raised against human ACE2 (Abcam, ab15348). Subsequent control sections were stained with Hematoxylin and Eosin (H&E) using standard procedures (**S1A–S1D Fig** in **S1 File**). All comparisons were made using sections that were stained in a single batch and imaged and processed using identical conditions (**S2 Fig** in **S1 File**).

## Image collection and quantification

Sections were imaged on a Nikon Eclipse E1000 using a Spot Insight 12 MP CMOS camera with the manufacturer's provided software. A single inflated field within each slide was selected using a 1x objective from which to start image collection by switching to the 20x objective. Using the 20x objective, 5 adjacent/sequential fields were collected. Images used for quantification were obtained by an investigator without knowledge of the patients' ages. ACE2 and cellular quantification were carried out using ImageJ (see **S Methods** in **S1 File**). The relationship of age to these measures was analyzed by simple linear regression using the *lm* function in R.

## Clinical feature comparison

Chart review was performed for each specimen from the ventilated cohort to determine the contribution of various features to ACE2 staining intensity. These parameters included: the date the sample was collected and preserved for archiving, the ventilation parameters used to support the patient closest in time to specimen collection (i.e., the fraction of inspired oxygen or $FiO_2$ and positive end-expiratory pressure or PEEP), the severity of DAD in the collected specimen graded by a board certified pathologist using the DAD Score [36] with minor modifications (see **S Methods** in **S1 File**), a clinical diagnosis of interstitial lung disease (ILD), a documented history of tobacco smoking, and patient sex. These features were compared to lung ACE2 staining using linear regression for continuous variables or the Wilcoxon ranksum test for categorical variables.

## Additional statistical analyses

Data were analyzed using R version 3.5.0 unless otherwise stated, the technical details are outlined above and as described in **S Methods in S1 File**. For multiple regression, no interaction terms were included. Outliers were not removed from any analysis. No power analysis was performed to determine the cohort sizes, sample size for the ventilated patient cohort was determined by tissue availability and feasibility of access. Pairwise comparisons were made using the Wilcoxon rank-sum test. A threshold of $p < 0.05$ was set for statistical significance, no correction for multiple testing was applied due to the limited number of statistical tests employed. All tests were two-tailed, between group comparisons.

## Results

To understand the age-related pattern of *ACE2* abundance in the human lung we utilized the large, harmonized RNAseq dataset available from the GTEx consortium [30]. GTEx version 8 contains data from 578 unique deceased donor lungs, binned by decade of age at death [33]. In unadjusted analysis, *ACE2* expression was not significantly associated with age ($\beta = 0.082$ TPM/decade, $p = 0.227$, linear regression) or sex ($p = 0.303$, Wilcoxon rank-sum test) (**Fig 1A and 1B**). The dataset also describes the mode of death, summarized by the Hardy scale [37], which categorizes clinical circumstances proximate to the end of life. The strongest category associated with *ACE2* expression was ventilation at the time of death showing a 103% increase in median *ACE2* levels (0.715 TPM without vs. 1.452 TPM with ventilation, $p < 1 \times 10^{-15}$, Wilcoxon rank-sum test). Lung *ACE2* expression, when stratified by Hardy score, revealed a significant relationship with age; expression of *ACE2* increased by 0.208 TPM for every decade of life ($p = 0.003$, multivariable regression), or a 14% increase each decade when compared to individuals in their 20's (**Fig 1C**). Transcript abundance from deceased donors can be affected by the tissue ischemic time that transpires prior to sample stabilization [38]. In order to assess whether this variable contributes to *ACE2* abundance from deceased donor lungs, we

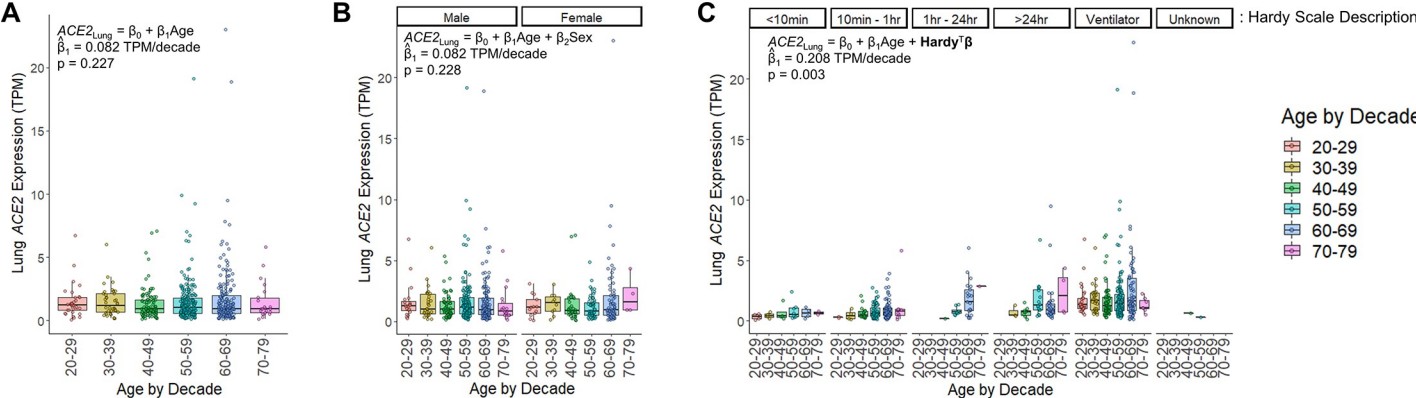

**Fig 1. *ACE2* RNA expression increases with age and mechanical ventilation in the lung. A)** Human lung *ACE2* expression is colored by decade of life when the individual died. In unadjusted linear regression, there was no significant effect of age on *ACE2* expression ($p = 0.227$, n = 578 individuals). **B)** *ACE2* expression in lung stratified by sex. Analysis adjusted for age and sex, revealed no significant effect for either variable ($p = 0.228$ for age, $p = 0.920$ for sex, multivariate regression, n = 395 males, n = 183 females). There was also no significant difference when compared by sex alone ($p = 0.303$, Wilcoxon rank-sum test). **C)** *ACE2* expression in lung stratified by the Hardy scale. This scale indicates the length of time spent in the terminal phase before death, which is depicted above each grouping of data points. Within each Hardy scale group, the data are sub-stratified by age. (n = 26 for a score of 1 representing a violent and fast death lasting <10 minutes, n = 156 for a score of 2 representing a fast death by natural causes lasting 10 minutes– 1 hour, n = 31 for a score of 3 representing an intermediate rate of death lasting 1 hour– 24 hours, n = 64 for a score of 4 representing a slow death with a terminal phase lasting > 24 hours, n = 299 for a score of 0 representing donors supported by a ventilator preceding death, n = 2 with an unknown score). For all panels, each point represents a sample from a unique individual. Box plots indicate quartiles. A linear model fit to the data is inset, indicating the estimated coefficient for age ($\beta_1$) and its significance.

incorporated this feature as a covariate in our model (**S3 Fig in S1 File**). The contribution of tissue ischemic time to *ACE2* expression was not significant (β = -0.0003 TPM/minute, p = 0.248) and did not change the effect of age on *ACE2* expression or its significance (β = 0.212 TPM/decade, p = 0.002). We also compared *ACE2* expression across the human body. Interestingly, when controlling for the manner of death, the lung exhibited the most significant increase in *ACE2* expression with age over all tissues (**S1 Table in S1 File**).

In order to characterize ACE2 protein abundance in the lung, we validated a commercial antibody directed against this key viral receptor (**S4A and S4B Fig in S1 File**). Prior analysis of ACE2 localization by IHC revealed strong enrichment within the luminal surface of cortical tubule cells of the kidney and enterocytes of the small intestine [39]. Our ACE2 IHC staining on healthy control tissue revealed a strikingly similar pattern, which highlighted the kidney cortex and specifically the apical membrane of brush border cells (**S4A Fig in S1 File**). We also detected strong ACE2 expression along villi and the apical membranes of small intestine enterocytes (**S4B Fig in S1 File**). Within the lung, ACE2 has been shown to localize to epithelial cells of the alveolus [39]. Data from murine scRNAseq reveal detectable expression of *Ace2* within type II pneumocytes (or alveolar type II / AT2 cells), ciliated columnar cells, alveolar macrophages, and to a lesser extent, stromal cells (**S5 Fig in S1 File**). Consistent with these data, our IHC revealed strong ACE2 staining within AT2 cells in normal alveoli, concentrated within the membrane, as well as alveolar macrophages (**Fig 2**). Taken together, these data indicate that our staining protocol accurately identifies ACE2 expressing cells within human tissue, including the lung. Therefore, we investigated whether the distribution of ACE2 in the lung is associated with ventilation and age.

We identified 12 lung samples from 11 patients requiring mechanical ventilation for AHRF available from our institution's archive. All 11 patients had DAD (**S2 Table in S1 File**), a pathological finding consistently observed in patients with severe Covid-19 lower respiratory disease [10, 11]. All tissue was collected prior to 2019 (range 2010–2018), excluding the possibility of SARS-CoV-2 involvement. H&E control sections revealed the expected histologic findings, depending on the phase of DAD, with reactive AT2 hyperplasia, hyaline membrane formation, interstitial thickening and/or fibrosis, and exudative edema (**S1B Fig in S1 File**). When stained by IHC, prominent ACE2 expression within the alveolar parenchyma was observed in reactive AT2 cells with increasing ACE2 staining intensity detected with advanced age (**Fig 3A and 3B**). ACE2 expression was quantified either by normalizing to tissue area (**Fig 3C**) or

Lung

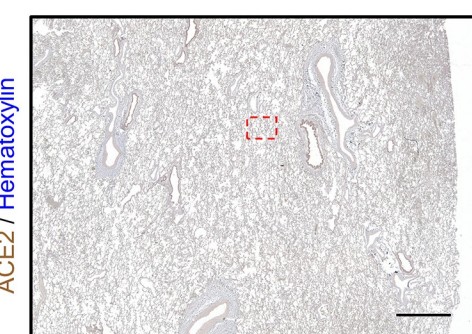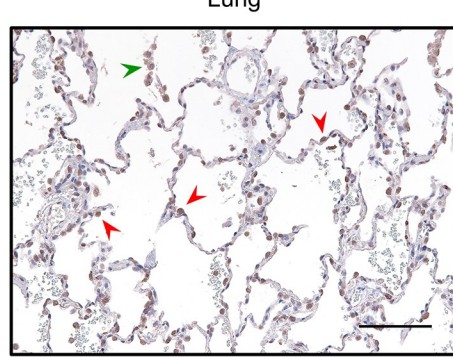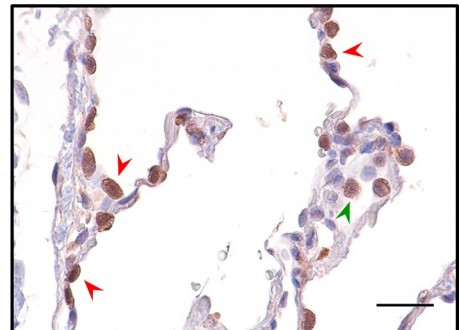

ACE2 / Hematoxylin

**Fig 2. IHC for ACE2 highlights AT2 cells and alveolar macrophages in normal lung.** In normal human lung from a 23-year-old female using a 1x objective (left, scale bar 3mm), strong ACE2 expression is observed within bronchioles and alveoli. The region outlined by the red dashed box is shown magnified at low power (middle, scale bar 200μm) which highlights prominent ACE2 expression in AT2 cells (red arrowheads) along the alveolar septum and in alveolar macrophages (green arrowhead). At high power (right, scale bar 50μm), ACE2 staining can be seen concentrated along the membrane of AT2 cells (red arrowheads) and within the cytoplasm of an alveolar macrophage (green arrowhead). Sections were stained for ACE2 using DAB and counterstained with hematoxylin.

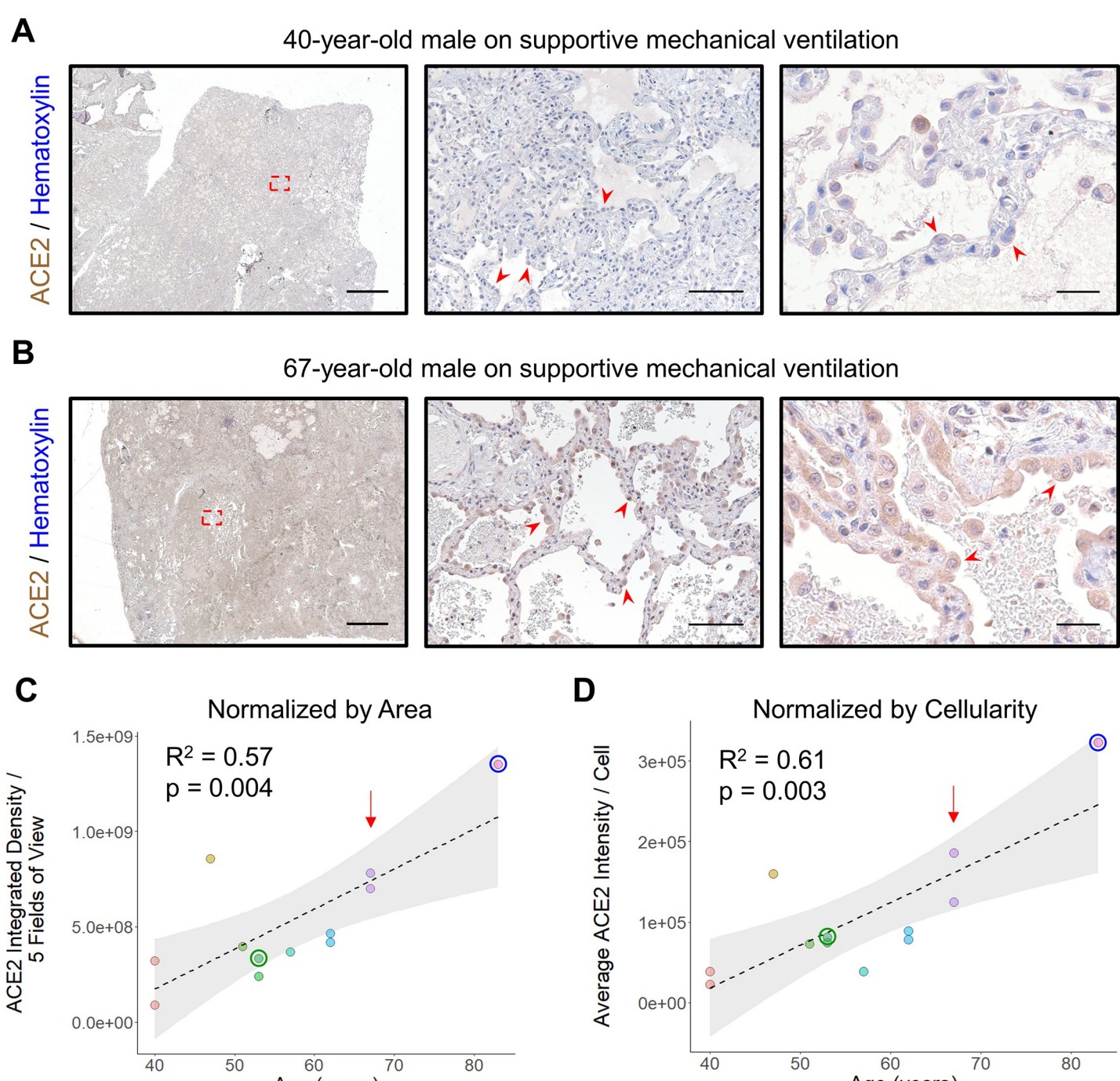

**Fig 3. ACE2 protein expression increases with age in ventilated patients. A)** Representative images of lung stained for ACE2 from a 40-year-old man with acute lung injury are shown using a 1x objective (left, scale bar 3mm) with the region outlined by the red dashed box magnified at low power (middle, scale bar 200μm) and a second field at high power (right, scale bar 50μm). Clusters of reactive AT2 cells (red arrowheads) are present along the alveolar septum which exhibit low level ACE2 expression. **B)** Representative images of lung stained for ACE2 from a 67-year-old man with acute lung injury superimposed on fibrosing interstitial lung disease are shown using a 1x objective (left, scale bar 3mm) with the region outlined by the red dashed box magnified at low power (middle, scale bar 200μm) and a second field at high power (right, scale bar 50μm). Numerous reactive AT2 cells (red arrowheads) exhibiting nucleomegaly and abundant cytoplasm can be seen demonstrating strong ACE2 staining. **C)** Quantitative IHC for ACE2 was carried out on samples from ventilated patients (n = 12 samples from 11 patients). Total ACE2 expression from 5 low power fields is plotted relative to the patient's age at the time of specimen collection. A linear fit to the data is indicated by the dashed line with the 95% confidence interval highlighted in grey. **D)** The same specimens quantitated in (C) were normalized by cellularity and the average ACE2 expression per cell is plotted along with a linear fit to the data and its 95% confidence interval. The red arrow in (C) and (D) indicates a patient providing 1 sample from the left lung and 1 sample from the right lung during the same procedure, utilized as a control for intra-individual reproducibility. The green and blue circle in (C) and (D) indicate the staining intensity of the samples depicted in Fig 5A and 5B, respectively. Sections were stained for ACE2 using DAB and counterstained with hematoxylin.

cellularity (**Fig 3D**) which revealed a significant increase with age (p = 0.004 and p = 0.003, respectively, linear regression). As a control for precision, 1 of the 11 patients had a double lung explant with 2 samples collected contemporaneously (1 from the left lung and 1 from the right lung). In this 67 year-old-man, the pathological findings were the same for each sample (**S2 Table in S1 File**). Importantly, both specimens yielded similar results when quantified for ACE2, indicating that our procedure measured its expression with good intra-individual reproducibility (**Fig 3C and 3D**). Excluding either of these samples did not change the effect of age on ACE2 expression (p = 0.009 by area and p = 0.003 by cellularity excluding the right lung; p = 0.008 by area and p = 0.006 by cellularity excluding the left lung, linear regression).

Alveolar macrophages comprised a minor fraction of total cells in samples from ventilated patients (median 3.4%, range 1.7% - 10.9%) and by visual inspection did not show the prominent age-related change in ACE2 expression seen in AT2 cells. To determine whether the abundance of alveolar macrophages contributes strongly to the effect of age on ACE2 expression, we quantified the number of alveolar macrophages in each sample. When normalized to either tissue area (**S6A Fig in S1 File**) or cellularity (**S6B Fig in S1 File**), the number of alveolar macrophages did not change with age (p = 0.768 and p = 0.427, respectively, linear regression).

We considered that factors other than age might be determining the intensity of ACE2 staining in lung samples from ventilated patients. Therefore, we studied a number of features including the length of time the sample was archived, the ventilation parameters used proximal to sample collection (namely $FiO_2$ and PEEP), the severity of DAD, the presence or absence of ILD, history of smoking, and patient sex. None of these features correlated significantly with ACE2 expression when normalized to either tissue area or cellularity (**Table 1** and S7A–S7D **Fig in S1 File**).

To compare our results in ventilated patients to samples collected from spontaneously breathing individuals, we collected an additional set of 20 lung samples from patients who underwent lung excision/biopsy for either pneumothorax repair or metastatic non-pulmonary neoplasms (**S2 Table in S1 File**). In all cases, care was taken to select portions of adjacent uninvolved lung parenchyma to analyze for ACE2 expression by IHC. Similar to our control sections, ACE2 staining was largely found within AT2 cells and alveolar macrophages (**Fig 4A and 4B**). When compared across ages, there was no significant difference either upon visual

**Table 1. Comparison of lung ACE2 expression with control features.**

| ACE2 Per Area (ACE2 Integrated Density / 5 Fields of View) | | | ACE2 Per Cell (ACE2 Integrated Density / Cell) | |
|---|---|---|---|---|
| Continuous Variables | | | | |
| Feature | Coefficient | p value | Coefficient | p value |
| Year Collected (per year) | 36842745 | 0.323 | 7544.049 | 0.408 |
| FiO2 (per percent) | -1417439 | 0.812 | -120.313 | 0.934 |
| PEEP (per cm H2O) | 24662684 | 0.408 | 8111.722 | 0.255 |
| DAD Score (per scale unit [0–24]) | 37546086 | 0.443 | 10003.1 | 0.399 |
| Categorical Variables | | | | |
| | p value | | p value | |
| ILD | 0.933 | | 0.933 | |
| History of Smoking | 1 | | 0.53 | |
| Sex | 0.482 | | 0.282 | |

ACE2 staining intensity was compared to various features derived from chart review of patients contributing samples to the ventilated cohort (n = 12 samples from 11 patients). For continuous variables a linear model was fit and the resulting β coefficient for that feature and its significance is indicated. For categorical variables the significance of a Wilcoxon rank-sum test is indicated.

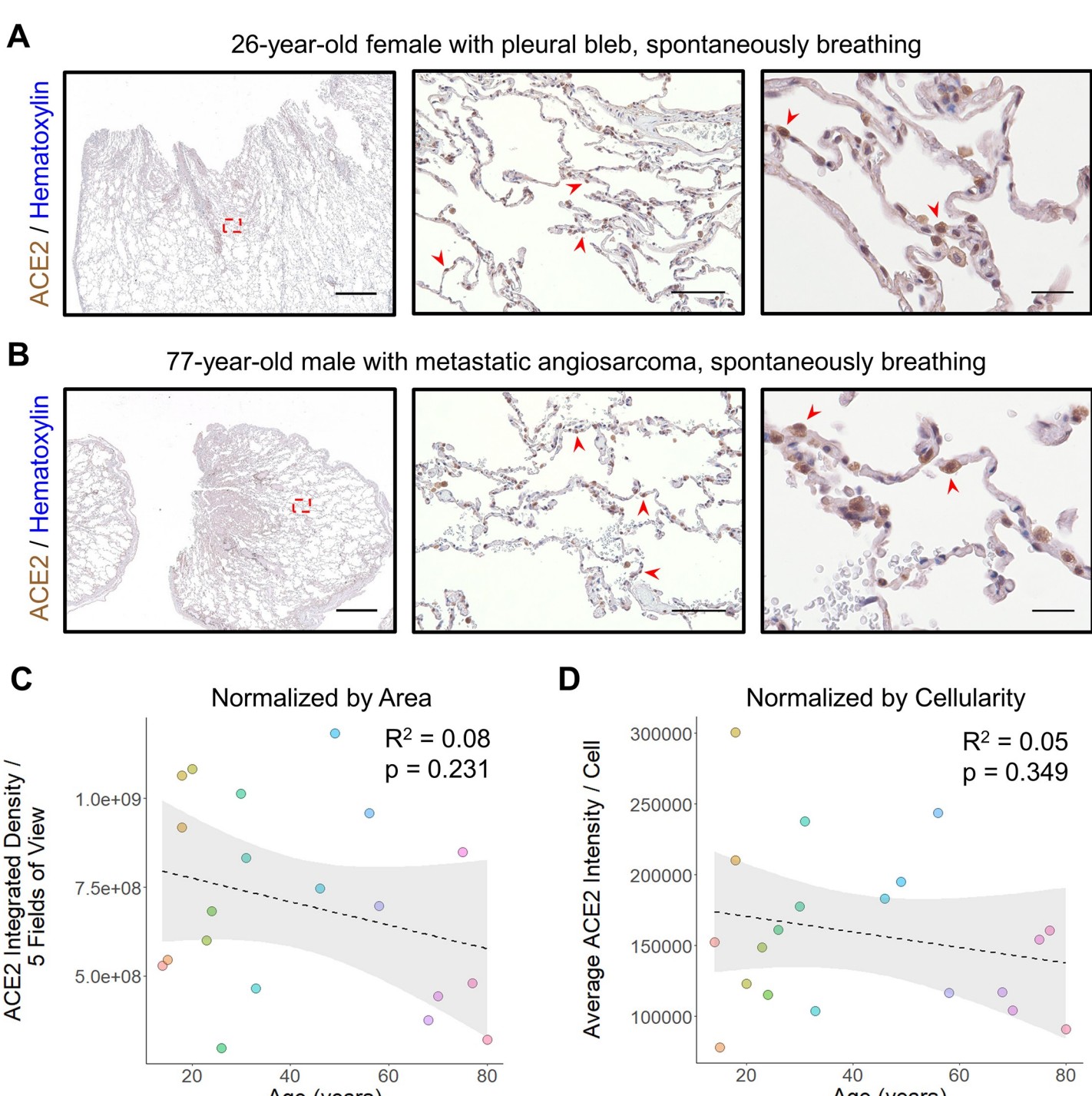

**Fig 4. ACE2 protein expression does not change with age in non-ventilated patients. A)** Representative images of lung stained for ACE2 from a 26-year-old female who underwent blebectomy for a spontaneous pneumothorax are shown using a 1x objective (left, scale bar 3mm) with the region outlined by the red dashed box magnified at low power (middle, scale bar 200μm) and a second field at high power (right, scale bar 50μm). The normal alveolated lung parenchyma reveals AT2 cells which are positive for ACE2 (red arrowheads) scattered among largely ACE2 low type I pneumocytes. **B)** Representative images of lung stained for ACE2 from a 77-year-old man undergoing wedge resection for suspected metastatic angiosarcoma are shown using a 1x objective (left, scale bar 3mm) with the region outlined by the red dashed box magnified at low power (middle, scale bar 200μm) and a second field at high power (right, scale bar 50μm). Histologic findings in the normal alveolated parenchyma adjacent to the lesion were similar to those found in (A) with strong ACE2 staining in scattered AT2 cells (red arrowheads) and occasional alveolar macrophages. **C)** Quantitative IHC for ACE2 was carried out on samples from non-ventilated patients (n = 20 patients). Total ACE2 expression from 5 low power fields is plotted relative to the patient's age at the time of specimen collection. A linear fit to the data is indicated by the dashed line with the 95% confidence interval highlighted in grey. **D)** The same specimens quantitated in (C) were normalized by cellularity and the average ACE2 expression per cell is plotted along with a linear fit to the data and its 95% confidence interval. Sections were stained for ACE2 using DAB and counterstained with hematoxylin.

inspection or when quantified normalizing to tissue area (p = 0.231, linear regression) (**Fig 4C**) or cellularity (p = 0.349, linear regression) (**Fig 4D**).

During our image acquisition, we noticed 2 cases exhibiting prominent ACE2 expression within the lung vascular endothelium (**Fig 5A and 5B**). This feature was not observed in any of the other 29 cases we examined. Upon chart review, we discovered that the first case was from a 53-year-old man who was actively receiving inpatient lisinopril at the time of lung explant (**Fig 5A**). The second case was from an 83-year-old man receiving valsartan during his immediately preceding admission at an outside institution, and at the time of transfer to our hospital, where his surgical resection occurred on post transfer day 4 (**Fig 5B**). Both patients were on supportive mechanical ventilation for AHRF prior to lung sample collection. The prominent age-related changes in AT2 cell ACE2 expression were again evident, but in addition, there was strong staining within the endothelium, indicating increased ACE2 expression within the vasculature. None of the other 29 cases in our dataset were from patients on angiotensin-converting enzyme inhibitors (ACEIs) or angiotensin receptor blockers (ARBs) (**S2 Table in S1 File**).

## Discussion

After the recognition that SARS-CoV-2 depends on ACE2 for host infection [23], a number of key studies have linked the expression of this viral receptor to tissue susceptibility. Within the

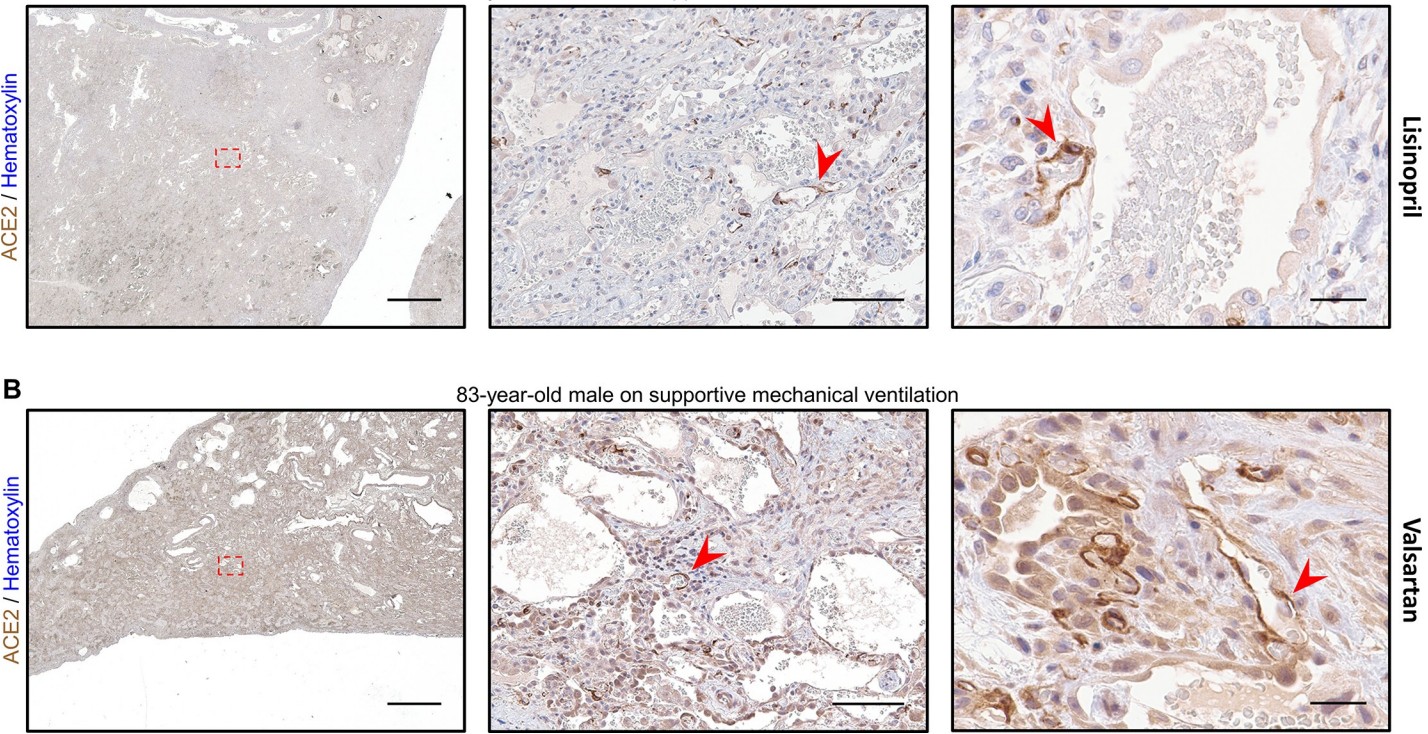

**Fig 5. Two patients on ACEI/ARB therapy exhibit intense endothelial ACE2 expression. A)** Representative images of lung stained for ACE2 from a 53-year-old man with organizing diffuse alveolar damage superimposed on fibrosing interstitial lung disease are shown using a 1x objective (left, scale bar 3mm) with the region outlined by the red dashed box magnified at low power (middle, scale bar 200μm) and second field at high power (right, scale bar 50μm). AT2 cell hyperplasia can be seen with lower level ACE2 expression present along the alveolar septum (green arrowheads). Strongly stained endothelial cells (red arrowheads) can be seen throughout the entire specimen from this patient who was on daily lisinopril at the time of specimen collection. **B)** Representative images of lung stained for ACE2 from an 83-year-old man in the organizing phase of diffuse alveolar damage are shown using a 1x objective (left, scale bar 3mm) with the region outlined by the red dashed box magnified at low power (middle, scale bar 200μm) and second field at high power (right, scale bar 50μm). In addition to high ACE2 expression in reactive AT2 cells (green arrowheads), strong staining can be seen in numerous vascular endothelial cells (red arrowheads) throughout the thickened septal area in this patient on valsartan 4 days prior to specimen collection. Sections were stained for ACE2 using DAB and counterstained with hematoxylin.

mammalian body, expression of *ACE2*, along with transmembrane serineprotease 2 (*TMPRSS2*)—the human protease responsible for activating the SARS-CoV-2 spike protein—coincides with cell types involved by Covid-19 [40]. ACE2 protein is highly expressed in differentiated gut enterocytes, leading to productive infection with SARS-CoV-2 [41]. Interestingly, *ACE2* expression is upregulated in the airway and lungs of smokers [29], a potential risk factor for severe disease [42]. Additionally, children have lower *ACE2* RNA expression in their nasal epithelium compared with adults [43]. A growing body of evidence suggests that ACE2 is highly associated with cellular susceptibility and disease severity [23, 44]. Given that the mechanism of death in Covid-19 typically involves severe lower respiratory tract infection, a disease feature strongly correlated with age, previous studies have sought to connect lung *ACE2* expression with aging [28, 29]. Although we analyzed a subset of the same publicly available data, we highlight the critical impact of mechanical ventilation on *ACE2* expression, an important physiologic change for the lung. By incorporating this feature into our model, we find that *ACE2* expression indeed increases in older individuals through what is likely a combination of advanced age, and either the need for or consequence of mechanical ventilation.

Reactive AT2 hyperplasia has long been recognized as a common finding after alveolar epithelial damage [45–48]. Patients in our ventilator supported cohort uniformly exhibited DAD with abundant hypertrophic AT2 cells. Our IHC analyses reveal that AT2 cells, as well as alveolar macrophages, are the primary cell types expressing ACE2 in the alveolus. Although we cannot rule out a subtle shift, changes in ACE2 expression among alveolar macrophages was minor when compared visually to AT2 cells and the former's abundance showed no relationship with age in ventilated patients. Rather, the age-related increase in ACE2 expression we observed appears to be primarily mediated by the level of this protein's expression in individual AT2 cells, with a potential lower contribution from other cells types. This trend was not seen in spontaneously breathing patients suggesting that the response of AT2 cells to alveolar injury drives this change.

Previous experiments have revealed that ACE2 expression has a special relationship with the cell cycle in AT2 cells responding to injury [49, 50]. For example, lung *ACE2* mRNA, protein, and enzymatic activity are dramatically reduced in patients with idiopathic pulmonary fibrosis (IPF), a type of chronic lung injury [49]. The same work revealed that these changes are produced in both rats and mice in response to experimental lung injury induced by bleomycin. A follow-up experiment revealed the underlying cellular mechanism for this response [50]. Similar to our results, a lung biopsy from a normal control lung revealed strong ACE2 expression in AT2 cells, which was lost in a patient with IPF coincident with the expression of PCNA in proliferating lung epithelial cells. Furthermore, A549 cells, a malignant human cell line of AT2 origin, express high levels of ACE2 when quiescent in culture, but rapidly reduce this expression upon cell cycle entry [50]. In the lungs of healthy control mice, stimulation of AT2 cell proliferation by keratinocyte growth factor was followed by the loss of ACE2 expression, which was restored 2 weeks later upon re-entry into quiescence [50]. Taken together these data reveal that ACE2 expression is ordinarily high in resting AT2 cells but can be lost upon mitotic entry driven by either injury or growth factor stimulation, contributing significantly to total lung ACE2 levels. This phenomenon could explain reduced expression in patients with DAD, but how might increased expression be seen in older patients who also exhibit AT2 hyperplasia in response to injury? A recent study has revealed that *ACE2* expression is upregulated in human respiratory cells in response to inflammatory signals including IFNα2 and IFNγ [40]. Immune changes associated with aging include the chronic, low-grade endogenous production of inflammatory cytokines, frequently referred to as "inflammaging", which is caused by a number of processes [51–54]. One potential model for the age-related differences we observe would be that in response to alveolar damage, AT2 cells enter the cell

cycle and ordinarily downregulate ACE2 expression. However, in older individuals, the increase in circulating inflammatory cytokines prevents this downregulation or even leads to ACE2 upregulation alongside AT2 cell proliferation. Alternatively, age associated epigenetic dysregulation may preclude the proper downregulation of ACE2 in proliferating AT2 cells [22]. Evidence for this second model derives from specific methylomic patterns, which shift during aging, leading to the proposal that these changes underlie an attenuated response to environmental cues in older organisms [55]. Indeed, in human airway epithelial cells, DNA methylation of the ACE2 promoter is decreased with age, predicting increased expression [56].

The above models imply that the relationship of aging to ACE2 expression is associated with the need for mechanical ventilation (i.e., the alveolar damage itself) instead of being caused by it. However, the reverse scenario also merits consideration. Positive pressure ventilation is associated with lung injury (ventilator-associated lung injury or VALI) and is sometimes the underlying cause of such injury (ventilator-induced lung injury or VILI) [57]. The pathological features of VILI are indistinguishable from other causes of ARDS and are induced by volume and pressure stress within the alveolus [58]. Ventilation strategies that result in VILI provoke higher levels of inflammatory cytokine release [59, 60]. In animal models, advanced age is associated with an increased likelihood of VILI [61] and exacerbated cytokine release in response to ventilation [62]. If lung ACE2 expression is triggered by mechanical ventilation itself via inflammatory cytokine production, then these factors may predispose older individuals to severe SARS-CoV-2 infection. The relative contributions of these and other potential mechanisms remain to be explored, however therapeutic strategies aimed at the underlying pathways could be envisioned. For example, reducing age associated inflammation may lower ACE2 expression in proliferating AT2 cells and limit infection of damaged alveolar epithelium. Indeed, the RECOVERY trial has revealed that dexamethasone reduces mortality in Covid-19 patients, with the greatest benefits achieved by those receiving mechanical ventilation [63]. The Janus kinase (JAK) pathway is downstream of many inflammatory cytokines associated with aging [64]. Results reported from a recent double-blind randomized control trial have shown that the JAK1/2 inhibitor, baricitinib, reduces time to recovery and improves outcomes in Covid-19 patients [65, 66]. Whether these therapeutics confer benefit, at least in part, by modulating ACE2 expression could be a point of future investigation.

Even though ACE2 expression correlates with cellular vulnerability to SARS-CoV-2 infection [23], decreased tissue levels are associated with physiologic worsening of acute lung injury (ALI) [67]. In mouse models of ALI, induced by either acid aspiration or sepsis, *Ace2* knockout results in more severe damage which can be rescued by intraperitoneal injection of recombinant ACE2 [68]. This effect appears to owe to increased levels of angiotensin II within the lung, as the severity of symptoms can be reversed by genetic disruption of *Ace* along with *Ace2*, and pharmacologic or genetic inhibition of angiotensin II signaling ameliorates lung injury. Interestingly, reduced ACE2 abundance in the lung can be triggered by SARS-CoV infection or intraperitoneal injection SARS-CoV spike protein alone [69]. Binding of SARS-CoV spike protein to ACE2 triggers its proteolytic cleavage by TNF-α-converting enzyme (TACE), providing a mechanism for this decrease [70]. The observed reduction in ACE2 levels leads to exacerbation of lung injury in mouse models. Thus ACE2 expression appears to confer susceptibility to viral infection at the cellular level yet protection from lung injury at the tissue level, how these opposing forces contribute to overall severity in the setting of SARS-CoV-2 infection requires additional study.

ACE2 expression alone likely cannot explain the cellular targets of SARS-CoV-2. Indeed, its spike protein diverges from SARS-CoV and the closely related bat coronavirus RaTG13 by the acquisition of the amino acids PRRA, conferring a furin cleavage site between the S1 and S2

subunits [71]. Cleavage at this site facilitates the adoption of an open conformation, exposing the receptor-binding domain of SARS-CoV-2 spike, and is provided by the ubiquitously expressed convertase furin [72]. The secondary cleavage of SARS-CoV-2 spike occurs at the S2' site which exposes the fusion peptide and leads to viral envelope merger with host cell membrane [73]. This proteolytic step is mediated primarily by TMPRSS2, although some priming may occur via cathepsin B and L [23]. The cellular distribution of TMPRSS2 is more restricted than furin, however *TMPRSS2* is highly expressed by AT2 cells [40]. Thus if productive infection requires all 3 proteins, the level of ACE2 expression may confer the crucial element limiting viral propagation in severe disease.

Although our study was originally designed to characterize the effects of age and ventilation on ACE2 expression, during data acquisition, the presence of strong vascular ACE2 staining in 2 cases was striking. This prompted us to investigate the underlying cause. Chart review uncovered that 1 patient was receiving an ACEI at the time of sample collection and the second was receiving an ARB just prior to collection. The remaining 29 cases were taken from patients not receiving either class of medication. ACE2 has been shown previously to be upregulated by ACEI/ARB therapy in the rodent heart [74] and human intestine [75], raising the concern that these therapeutics may predispose to SARS-CoV-2 infection [76]. Observational studies have indicated that the use of ACEI/ARBs does not increase the chances of testing positive for SARS-CoV-2 [77] and reduces the chances of Covid-19 mortality [78]. Randomized trials to assess the impact of ACEI/ARB discontinuation in Covid-19 are currently planned and ongoing [79, 80]. Although we identified elevated pulmonary endothelial ACE2 expression in the only 2 patients on ACEI/ARBs in our cohort, this finding is preliminary and requires replication. A recent study has demonstrated pulmonary vascular endothelialitis associated with SARS-CoV-2 infection of these cells [9]. Whether increased endothelial ACE2 expression modulates this disease feature or Covid-19 pathogenesis in general is currently unknown.

## Limitations

In this small series of cases, we identified increased *ACE2* RNA and ACE2 protein expression within the human lung associated with age, when controlling for ventilator status. Although our RNAseq analyses drew on the sample rich resource of the GTEx dataset, including 578 unique donor lung samples, our quantitative IHC results derived from limited archival samples (12 samples from 11 patients). However, the concordance between our findings at the RNA and protein level revealing an interaction of mechanical ventilation and age in both cases, strengthens our conclusions.

## Supporting information

**S1 File.**
(DOCX)

## Acknowledgments

We would like to thank the staff of the Stanford Medicine Department of Pathology for excellence in patient care throughout the current pandemic and selfless diligence in cataloguing, processing, and reporting the original case material contributing to this study. We also extend our sincere appreciation to Norman Cyr for assistance with image acquisition and processing and Erna Forgo for help with image acquisition and initial interpretation. This manuscript has been released as a pre-print at medRxiv, medRxiv 2020.07.05.20140467 [81].

## Author Contributions

**Conceptualization:** Steven Andrew Baker, Gerald J. Berry, Thomas J. Montine.

**Formal analysis:** Steven Andrew Baker.

**Funding acquisition:** Gerald J. Berry, Thomas J. Montine.

**Investigation:** Steven Andrew Baker, Shirley Kwok, Gerald J. Berry, Thomas J. Montine.

**Methodology:** Steven Andrew Baker, Gerald J. Berry, Thomas J. Montine.

**Resources:** Gerald J. Berry, Thomas J. Montine.

**Supervision:** Gerald J. Berry, Thomas J. Montine.

**Validation:** Steven Andrew Baker.

**Writing – original draft:** Steven Andrew Baker.

**Writing – review & editing:** Steven Andrew Baker, Gerald J. Berry, Thomas J. Montine.

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
