## [Decision Letter · Decision Letter 0]

10 Nov 2020

PONE-D-20-32785

Angiotensin-converting enzyme 2 (ACE2) expression increases with age in patients requiring mechanical ventilation.

PLOS ONE

Dear Dr. Baker,

Thank you for submitting your manuscript to PLOS ONE. After careful consideration, we feel that it has merit but does not fully meet PLOS ONE’s publication criteria as it currently stands. Therefore, we invite you to submit a revised version of the manuscript that addresses the points raised during the review process.

The manuscript by Baker et al. is well assessed by two expert reviewers; however, it does not reach to an enough level for the acceptance in PlosOne in the present form. 

See the Reviewers' comments carefully and respond them appropriately.

We look forward to receiving your revised manuscript.

Kind regards,

Masaki Mogi

Academic Editor

PLOS ONE

Journal Requirements:

2.  Thank you for including your ethics statement: 'Formalin-fixed paraffin-embedded (FFPE) human lung excisions/biopsies received through Stanford Surgical Pathology during 2010-2020 were selected in accordance with an approved IRB protocol.'     

(a) Please amend your current ethics statement to include the full name of the ethics committee/institutional review board(s) that approved your specific study.  

(b) Once you have amended this/these statement(s) in the Methods section of the manuscript, please add the same text to the “Ethics Statement” field of the submission form (via “Edit Submission”).

3. In your ethics statement in the manuscript and in the online submission form, please provide additional information about the patient samples used in your retrospective study. Specifically, please ensure that you have discussed whether all data were fully anonymized before you accessed them.

4. In the Methods section, please provide the accession number of the RNA-seq dataset analyzed in your study.

5. To comply with PLOS ONE submission guidelines, in your Methods section, please provide additional information regarding your statistical analyses. For more information on PLOS ONE's expectations for statistical reporting, please see https://journals.plos.org/plosone/s/submission-guidelines.#loc-statistical-reporting.

6. We noted in your submission details that a portion of your manuscript may have been presented or published elsewhere. [

"Yes, the work was published as a pre-print on medRxiv. Please see:

" ext-link-type="uri" xlink:type="simple">https://doi.org/10.1101/2020.07.05.20140467"

Please clarify whether this publication was peer-reviewed and formally published. If this work was previously peer-reviewed and published, in the cover letter please provide the reason that this work does not constitute dual publication and should be included in the current manuscript.

Reviewers' comments:

Reviewer's Responses to Questions

**Comments to the Author**

1. Is the manuscript technically sound, and do the data support the conclusions?

Reviewer #1: Yes

Reviewer #2: Partly

2. Has the statistical analysis been performed appropriately and rigorously? 

Reviewer #1: Yes

Reviewer #2: Yes

3. Have the authors made all data underlying the findings in their manuscript fully available?

Reviewer #1: Yes

Reviewer #2: Yes

4. Is the manuscript presented in an intelligible fashion and written in standard English?

Reviewer #1: Yes

Reviewer #2: Yes

5. Review Comments to the Author

Reviewer #1: Authors showed in this obsevation study, ACE2 expression increases with age in the setting of alveolar damage observed in patients on mechanical ventilation, providing a potential mechanism for higher Covid-19 mortality in the elderly. This is unique study to know Covid-19 infection. However, authors did not clearly show that how we manage these patient to recover. So authors should provide some solution, for example, if clinician face to the Covid-19 patient how they manage this disease.

Reviewer #2: Baker et al. found that ACE2 expression increased with age in lungs of patients requiring mechanical ventilation. To elucidate the mechanism of increased Covid-19 mortality in the elderly, the authors analyzed ACE2 expression of lungs using publicly available RNA sequencing data, GTEx, and immunohistochemistry (IHC) of the postmortem specimens. Analysis of GTEx reveals expression of ACE2 is upregulated with age in ventilator group. From IHC analysis, the authors found ACE2 was strongly upregulated with age in the ventilator group, associated with prominent expression in AT2 cells. On the other hand, ACE2 expression did not change with age in spontaneously breathing group in both GTEx and IHC analysis. The study is well written and open new insight on the relationship between aging and ACE2 expression in lung.

Major issues

1. Regarding the discussion, I think that the mechanism of increased expression of ACE2 in elder ventilator group is one of the main points of this paper and was well considered, but it is difficult to understand in this description. In particular, it is difficult to understand the relationship between cell cycle of AT2 cells and expression of ACE2, and the prevention of ACE2 down-regulation due to epigenetic changes in elderly, so it is better to supplement the contents of citations 47 and 22.

2. The authors concluded that “ACE2 expression increases with age in the setting of alveolar damage observed in patients on mechanical ventilation, providing a potential mechanism for higher Covid-19 mortality in the elderly”. Indeed, increased expression of ACE2 in the lung has been shown to correlate with prevalence and exacerbation of Covid-19. But at the same time, animal studies have shown that ACE2 acts protectively on ARDS (1). I think you need to mention and add some discussion about the protective role of ACE2 in respiratory disease.

(1) Imai Y, Kuba K, Rao S, Huan Y, Guo F, Guan B, Yang P, Sarao R, Wada T, Leong-Poi H, Crackower MA, Fukamizu A, Hui CC, Hein L, Uhlig S, Slutsky AS, Jiang C, Penninger JM. Angiotensin-converting enzyme 2 protects from severe acute lung failure. Nature. 2005;436:112-116

3. The results of this study may suggest that ACE2 expression increases with age in patients with DAD, as all samples used as a ventilator group in this study had DAD. Is there data on the correlation between the severity of DAD and age-related changes in ACE2?

4. The results of increased expression of ACE2 in the ACE inhibitor or ARB administration individuals are interesting, but these are preliminary data. It might be better to avoid referencing it in abstract. Then, I would like to know the ACE2 staining intensity of these samples to compare them with other samples. Please add arrows indicating the staining intensity of these two samples to Fig3 C and D.

5. The authors need to mention not only TMPRSS2 but also Furin as protease involved in the activation of the SARS-CoV2 spike protein in the discussion. And also need to add discussion about TACE, the negative regulator of ACE2 expression.

Minor concerns

1. In Figure 2-5, could the authors include images of the entire lung and insets with the area of interest?

2. Please arrange the lung-stained images of the ventilator and spontaneous breathing group by age.

3. Figure 2 A and B should be moved to supplementary materials. I think these are positive control for ACE2 staining, not the main result.

4. Please add references to “A growing body of evidence suggests that ACE2 is highly associated with disease susceptibility and severity”. (p9, line 4-5)

6. PLOS authors have the option to publish the peer review history of their article (what does this mean?). If published, this will include your full peer review and any attached files.

Reviewer #1: No

Reviewer #2: No

---

## [Author Response · Author response to Decision Letter 0]

15 Jan 2021

We are grateful to both of the reviewers and the academic editor for making many excellent suggestions that we integrated into the manuscript and addressed with additional analyses. The new data provide much stronger support for the conclusions of the study and place our findings better within the context of the rich literature existing from investigators in the field. Below is an item-by-item response to all the suggestions. Reviewers’ comments are in blue font for ease of reference.

Reviewer #1: Authors showed in this obsevation study, ACE2 expression increases with age in the setting of alveolar damage observed in patients on mechanical ventilation, providing a potential mechanism for higher Covid-19 mortality in the elderly. This is unique study to know Covid-19 infection. However, authors did not clearly show that how we manage these patient to recover. So authors should provide some solution, for example, if clinician face to the Covid-19 patient how they manage this disease.

Thank you kindly for your review and extremely helpful suggestion. In addition to the direct viral RNA polymerase inhibitor (remdesivir), anti-inflammatory medications have emerged as a cornerstone in the treatment of severe Covid-19, particularly dexamethasone. Additionally, unpublished randomized control data announced by Eli Lilly (from the ACTT-2 trial) have revealed that the JAK1 and JAK2 inhibitor baricitinib improves outcomes in patients with Covid-19. At first, this might seem counterintuitive as the FDA lists severe viral infection requiring hospitalization as a contraindication to baricitinib (https://www.accessdata.fda.gov/drugsatfda_docs/label/2018/207924s000lbl.pdf). However, ACE2 expression has been demonstrated to be downstream of IFNα2 and IFNγ (Ziegler et al.; ref. 40 in our manuscript) and these two cytokines signal through JAK1/2 (please see Rönnblom and Leonard Lupus Sci Med 2019, PMID: 31497305 for an excellent review). We hypothesize that the potentially counterintuitive effect of anti-inflammatory medications might be working by reducing ACE2 expression in damaged lung. We have now included a discussion of these potential therapeutics in the context of our findings (please see lines 294-302). We are very grateful that you motivated our discussion of this idea.

Reviewer #2: Baker et al. found that ACE2 expression increased with age in lungs of patients requiring mechanical ventilation. To elucidate the mechanism of increased Covid-19 mortality in the elderly, the authors analyzed ACE2 expression of lungs using publicly available RNA sequencing data, GTEx, and immunohistochemistry (IHC) of the postmortem specimens. Analysis of GTEx reveals expression of ACE2 is upregulated with age in ventilator group. From IHC analysis, the authors found ACE2 was strongly upregulated with age in the ventilator group, associated with prominent expression in AT2 cells. On the other hand, ACE2 expression did not change with age in spontaneously breathing group in both GTEx and IHC analysis. The study is well written and open new insight on the relationship between aging and ACE2 expression in lung.

Thank you so much.

Major issues

1. Regarding the discussion, I think that the mechanism of increased expression of ACE2 in elder ventilator group is one of the main points of this paper and was well considered, but it is difficult to understand in this description. In particular, it is difficult to understand the relationship between cell cycle of AT2 cells and expression of ACE2, and the prevention of ACE2 down-regulation due to epigenetic changes in elderly, so it is better to supplement the contents of citations 47 and 22.

This is an excellent point and we have now expanded our discussion of these two important ideas from the literature. We have highlighted the work of Li et al. (ref 49) and Uhal et al. (ref 50) to further elaborate on the mitotic cycle dependence of AT2 cell ACE2 expression (please see lines 254-269). We have also more explicitly addressed the evidence concerning the epigenetic regulation of ACE2 expression with age, particularly in the lung (please see lines 277-282).

2. The authors concluded that “ACE2 expression increases with age in the setting of alveolar damage observed in patients on mechanical ventilation, providing a potential mechanism for higher Covid-19 mortality in the elderly”. Indeed, increased expression of ACE2 in the lung has been shown to correlate with prevalence and exacerbation of Covid-19. But at the same time, animal studies have shown that ACE2 acts protectively on ARDS (1). I think you need to mention and add some discussion about the protective role of ACE2 in respiratory disease.

(1) Imai Y, Kuba K, Rao S, Huan Y, Guo F, Guan B, Yang P, Sarao R, Wada T, Leong-Poi H, Crackower MA, Fukamizu A, Hui CC, Hein L, Uhlig S, Slutsky AS, Jiang C, Penninger JM. Angiotensin-converting enzyme 2 protects from severe acute lung failure. Nature. 2005;436:112-116

We completely agree, the role of ACE2 in lung injury is a rich topic that we have now discussed in further detail. Although we believe that increased ACE2 expression in the elderly may, on balance, be deleterious in Covid-19, as the receptor for SARS-CoV-2, it may also have both evolutionary and empirical benefits for lung physiology. We have included the important work of Imai et al. (ref. 68) and Kuba et al. (ref 69), as well as other studies in the Discussion (please see lines 303-315).

3. The results of this study may suggest that ACE2 expression increases with age in patients with DAD, as all samples used as a ventilator group in this study had DAD. Is there data on the correlation between the severity of DAD and age-related changes in ACE2?

Thanks again for this point. This is a question we also had and addressed preliminarily but had not included in the manuscript. Given your intuitive question, we have now added new data to investigate this and other potential confounders of increased ACE2 expression with age (please see the new Table 1 and S7 Fig). In brief, we found no significant relationship between DAD severity and ACE2 expression. We also did not find any effect of the length of time samples were stored after collection, ventilation parameters used to support the patients, the presence or absence of ILD, a history of smoking, or patient sex upon ACE2 expression. We have included a description of these analyses in the Methods (please see lines 117-126) and Results (please see lines 195-205). 

4. The results of increased expression of ACE2 in the ACE inhibitor or ARB administration individuals are interesting, but these are preliminary data. It might be better to avoid referencing it in abstract. Then, I would like to know the ACE2 staining intensity of these samples to compare them with other samples. Please add arrows indicating the staining intensity of these two samples to Fig3 C and D.

Certainly, we have now removed that sentence from the abstract (thanks!). We have circled the data points in Fig 3C and 3D corresponding to the staining intensity of the samples from Fig 5 (please see lines 660-661).

5. The authors need to mention not only TMPRSS2 but also Furin as protease involved in the activation of the SARS-CoV2 spike protein in the discussion. And also need to add discussion about TACE, the negative regulator of ACE2 expression.

Agreed, we have added these points which help clarify the importance of all 3 proteins. Please see lines 316-326 for a discussion of TMPRSS2 and Furin expression and lines 309-315 for a discussion of TACE.

Minor concerns

1. In Figure 2-5, could the authors include images of the entire lung and insets with the area of interest?

Thanks, we have now included the original 1x image for all figures and indicated the field that this magnified in the adjacent subpanel.

2. Please arrange the lung-stained images of the ventilator and spontaneous breathing group by age.

Corrected, we believe this makes the data more easily interpretable in chronological order, thanks!

3. Figure 2 A and B should be moved to supplementary materials. I think these are positive control for ACE2 staining, not the main result.

They have been moved to the supplement (please see S4 Fig).

4. Please add references to “A growing body of evidence suggests that ACE2 is highly associated with disease susceptibility and severity”. (p9, line 4-5)

Again an excellent point, we have clarified our intended meaning of the sentence as “A growing body of evidence suggests that ACE2 is highly associated with cellular susceptibility and disease severity”, and referenced the work of Hoffmann et al. (ref. 23) demonstrating the cell type specificity that ACE2 expression confers for SARS-CoV-2 infection and Pinto et al. (ref. 44) demonstrating that lung ACE2 RNA expression is upregulated in comorbid conditions known to predispose patients to severe Covid-19. Thanks so much for helping us clarify this sentence.

Response to the Academic Editor

Thank you so much for providing the links to the correct formatting options, we have now updated the manuscript to the best of our knowledge.

2. Thank you for including your ethics statement: 'Formalin-fixed paraffin-embedded (FFPE) human lung excisions/biopsies received through Stanford Surgical Pathology during 2010-2020 were selected in accordance with an approved IRB protocol.' 

(a) Please amend your current ethics statement to include the full name of the ethics committee/institutional review board(s) that approved your specific study.

We have amended our ethics statement in the Methods sections to include full name and reference to our IRB protocol (Stanford University IRB # 33727). Please see lines 97-98.

(b) Once you have amended this/these statement(s) in the Methods section of the manuscript, please add the same text to the “Ethics Statement” field of the submission form (via “Edit Submission”).

We will make this amendment as you suggested, thank you.

3. In your ethics statement in the manuscript and in the online submission form, please provide additional information about the patient samples used in your retrospective study. Specifically, please ensure that you have discussed whether all data were fully anonymized before you accessed them.

Although the investigator performing the data acquisition and statistical analyses (SAB) was blinded to patient demographics during the collection phase, access to the medical chart, in accordance with our IRB protocol, was used to perform chart review. No identifying information is included in the manuscript.

4. In the Methods section, please provide the accession number of the RNA-seq dataset analyzed in your study.

Thank you we have now included this accession ID (dbGaP accession phs000424.v8.p2). Please see line 84.

5. To comply with PLOS ONE submission guidelines, in your Methods section, please provide additional information regarding your statistical analyses. For more information on PLOS ONE's expectations for statistical reporting, please see https://journals.plos.org/plosone/s/submission-guidelines.#loc-statistical-reporting.

 We have updated the Additional Statistical Analysis section to include the information described in the guidelines (please see lines 127-134).

6. We noted in your submission details that a portion of your manuscript may have been presented or published elsewhere. [

"Yes, the work was published as a pre-print on medRxiv. Please see:

https://doi.org/10.1101/2020.07.05.20140467"

Please clarify whether this publication was peer-reviewed and formally published. If this work was previously peer-reviewed and published, in the cover letter please provide the reason that this work does not constitute dual publication and should be included in the current manuscript.

The pre-print was published on medRxiv and was not peer-reviewed.

Thanks, we have now included these at the end of the manuscript (please see lines 699-709).

---

## [Decision Letter · Decision Letter 1]

1 Feb 2021

Angiotensin-converting enzyme 2 (ACE2) expression increases with age in patients requiring mechanical ventilation.

PONE-D-20-32785R1

Dear Dr. Baker,

We’re pleased to inform you that your manuscript has been judged scientifically suitable for publication and will be formally accepted for publication once it meets all outstanding technical requirements.

Kind regards,

Masaki Mogi

Academic Editor

PLOS ONE

Additional Editor Comments (optional):

The authors have well response to the reviewers' comments. No further comments.

Reviewers' comments:

Reviewer's Responses to Questions

**Comments to the Author**

1. If the authors have adequately addressed your comments raised in a previous round of review and you feel that this manuscript is now acceptable for publication, you may indicate that here to bypass the “Comments to the Author” section, enter your conflict of interest statement in the “Confidential to Editor” section, and submit your "Accept" recommendation.

Reviewer #1: All comments have been addressed

Reviewer #2: All comments have been addressed

2. Is the manuscript technically sound, and do the data support the conclusions?

Reviewer #1: Yes

Reviewer #2: Yes

3. Has the statistical analysis been performed appropriately and rigorously? 

Reviewer #1: Yes

Reviewer #2: Yes

4. Have the authors made all data underlying the findings in their manuscript fully available?

Reviewer #1: Yes

Reviewer #2: Yes

5. Is the manuscript presented in an intelligible fashion and written in standard English?

Reviewer #1: Yes

Reviewer #2: Yes

6. Review Comments to the Author

Reviewer #1: Thank you kindly for your review and extremely helpful suggestion. In addition to the direct viral RNA polymerase inhibitor (remdesivir), anti-inflammatory medications have emerged as a cornerstone in the treatment of severe Covid-19, particularly dexamethasone. Additionally, unpublished randomized control data announced by Eli Lilly (from the ACTT-2 trial) have revealed that the JAK1 and JAK2 inhibitor baricitinib improves outcomes in patients with Covid-19. At first, this might seem counterintuitive as the FDA lists severe viral infection requiring hospitalization as a contraindication to baricitinib (https://www.accessdata.fda.gov/drugsatfda_docs/label/2018/207924s000lbl.pdf). However, ACE2 expression has been demonstrated to be downstream of IFNα2 and IFNγ (Ziegler et al.; ref. 40 in our manuscript) and these two cytokines signal through JAK1/2 (please see Rönnblom and Leonard Lupus Sci Med 2019, PMID: 31497305 for an excellent review). We hypothesize that the potentially counterintuitive effect of anti-inflammatory medications might be working by reducing ACE2 expression in damaged lung. We have now included a discussion of these potential therapeutics in the context of our findings (please see lines 294-302). We are very grateful that you motivated our discussion of this idea.

Your answer to the above question is accurate.

Reviewer #2: The author responded politely to all questions and the manuscript has been improved very much.

No further revision is required.

7. PLOS authors have the option to publish the peer review history of their article (what does this mean?). If published, this will include your full peer review and any attached files.

Reviewer #1: No

Reviewer #2: No

---

## [Editor Report · Acceptance letter]

5 Feb 2021

PONE-D-20-32785R1 

Angiotensin-converting enzyme 2 (ACE2) expression increases with age in patients requiring mechanical ventilation. 

Dear Dr. Baker:

I'm pleased to inform you that your manuscript has been deemed suitable for publication in PLOS ONE. Congratulations! Your manuscript is now with our production department. 

Kind regards, 

on behalf of

Dr. Masaki Mogi 

Academic Editor

PLOS ONE